# Determining Magnitudes of Forces at Known Locations through a Strain Gauge Force Transducer

**DOI:** 10.3390/s23167017

**Published:** 2023-08-08

**Authors:** Edward Bednarz, Christian Dietrich, Brad Hepner, Jay Patel, Abas Sabouni

**Affiliations:** Department of Mechanical and Electrical Engineering, Wilkes University, Wilkes-Barre, PA 18766, USA; christian.dietrich@wilkes.edu (C.D.); brad.hepner@wilkes.edu (B.H.J.); jay.patel4@wilkes.edu (J.P.); abas.sabouni@wilkes.edu (A.S.)

**Keywords:** strain gauge, load identification, force transducer, structural analysis

## Abstract

A novel strain gauge force transducer was developed to minimize the number of strain gauges needed to determine the magnitudes of loads when the locations are known. This innovative methodology requires only one strain gauge for each force magnitude desired, reducing the complexity and cost associated with traditional approaches. The theory was verified with laboratory experiments. Seven uniaxial strain gauges were attached to the underside of a simply supported, slender, aluminum beam. One or more loads were applied either directly atop strain gauges or in known positions between strain gauges. Experiments were conducted on several different single and double-load configurations to evaluate the extent of the new methodology which yielded average errors under 5% for the cases where loads were direct atop strain gauges and 6.6% for the cases where the loads were between strain gauges. These findings indicate the potential of this novel strain gauge force transducer to revolutionize load measurement in scenarios where load locations are predetermined.

## 1. Introduction

Strain gauge force transducers are not new in the realm of engineering, as strain gauges are especially useful and prominent tools in structural analysis. This methodology implements the use of strain gauges on a small-scale laboratory experiment with the ability to expand as needed into larger structures. A transducer is an instrument that converts one form of energy into another [1].

Regarding the context of the laboratory experiment, the mechanical energy of the applied forces are converted into an electrical signal via the use of a Wheatstone bridge circuit that is integrated in a Vishay P3 recorder [2]. Strain gauges are constructed on a thin, flexible material with high-gauge wire. These gauges operate such that the resistance of the wire changes under an applied load. The Wheatstone bridge that is used is able to measure very fine changes in the resistance of the wire. Essentially, the circuit is made up of four resistors wired in a diamond pattern, with three of those resistors being highly precise values and the other being the strain gauge in the system to complete the circuit. By assessing the change of resistance of the strain gauge, the strain value is the response to an applied load.Load identification and the determination of forces and/or loads on a structure from measured responses are critical in monitoring structural health. The process as a whole is challenging due to the uncertainty of measurements and the complexity of loads.

As these tools are very effective, they have increasingly become a topic of research for load determination. An alternative method of force determination to strain gauges are Fiber Bragg Grating (FBG) sensors and fiber optics, which are prominently used in structural analysis. One study shows that researchers were able to create a method to identify the load of a moving car on a bridge through the use of long-gauge fiber optic strain responses [3]. Similarly, another study shows that the use of an FBG sensor is able to measure longitudinal strains when a beam is subjected to multiple loads [4]. An additional study looked at using an FGB sensor to measure the force and position of a weight that was attached to a surgical forceps [5]. The practicality of FBG sensors continues as these are also used in measuring dynamic strain in cantilever beams, which can yield strains in linear and nonlinear systems [6]. Ultimately, the decision was made to focus on strain gauges. Strain gauges display remarkable accuracy in the detection of minor changes of resistance which as a result can precisely quantify any applied loads. The methodology found in this study reduces the number of necessary strain gauges and promises a more cost-effective approach which results in a practical application of identifying loads.

Innovative techniques using nonlinear strain gauges are frequently being studied, such as one that highlights a triaxial force transducer and the calibration method [7]. Although rigidly mounted, the strain gauges are still able to observe dynamic forces such as those caused by acceleration and vibration, thus making this method highly desirable in load testing for aircraft and automobiles [7]. As the testing highlighted in this study is static, nonlinear strain gauges are not necessary. Furthermore, the strain present is only in one direction, thus making a triaxial strain gauge excessive.

Previously, a strain gauge force transducer was created to determine both the magnitudes and locations of several applied forces at the same time [8]. To do so, four strain gauges are required for the initial force and two more are necessary for each additional applied force. The methodology focused on in this study aimed to reduce the number of necessary strain gauges for determining the load magnitude by eliminating the unknown locations. The benefit of knowing the location where the force is applied is such that only one strain gauge is needed per force.

## 2. Theory

Consider a simply supported beam with a given length *L* subjected to a force *F* (see Appendix A) at a location *A*, and that has two reaction forces at each end, $R_{A}$ and $R_{B}$, as shown in Figure 1. The beam is assumed to be in static equilibrium. Therefore, it is expected that the maximum moment, $M_{\max}$ is to occur at the point of the applied force *A*, as shown in Figure 2. To recreate this graph, a piece-wise function composed of two equations is used: one representing the increasing moment before the point of the applied force and the other representing the decreasing moment after the point of the applied force [9].
(1)$$M=\left\{\begin{array}{cc} Fx(1-\frac{A}{L}) & 0\leq x\leq A\\ FA(1-\frac{x}{L}) & A\leq x\leq L \end{array}\right.$$

The applied force *F* can be determined if the moment at the location of the force *A* is known, from Equation (Equation 1). However, it is not possible to directly measure the moment, so the relationship between bending stress and moment must be utilized. This relationship is represented in Equation (Equation 2), where σ denotes bending stress, *y* is the distance from the neutral axis of the beam to where the stress is calculated, and *I* is the area moment of inertia [10].
(2)$$\sigma =-\frac{My}{I}$$

Bending stress itself cannot be directly measured, but it can be accurately calculated using the well-established relationship between stress and strain. This relationship is represented in Equation (Equation 3), where *E* denotes Young’s modulus, a material property that quantifies the stiffness of a solid, and ε symbolizes the strain, a dimensionless measure of the deformation experienced by the material. By utilizing this fundamental relationship, it is possible to infer the bending stress from the measured strain values [10].
(3)σ=Eε

Strain, being a physical quantity, can be measured using strain gauges. By combining Equations (Equation 2) and (Equation 3), a relationship can be established between the bending moment and strain resulting in Equation (Equation 4).
(4)$$M=-\frac{EI}{y}\varepsilon$$

A calibration factor, β, is used to account for different geometric and material properties. In this experiment, one β was used since a beam that was homogeneous with a constant cross-section was utilized. However, if the beam has variable width or thickness, or is non-homogeneous, β must be calculated separately for each strain gauge. The experimental β can be determined with the use of a known force value and a tool such as Excel Solver. The theoretical β can be represented by Equation (Equation 5). Equation (Equation 4) was subsequently rewritten to include β and is represented by Equation (Equation 6).
(5)$$\beta =-\frac{EI}{y}$$
(6)M=βε

By using the relationship established between moment and strain, the magnitude of a force can be calculated using Equations (Equation 1) and (Equation 6). In situations where multiple forces are present, as shown in Figure 3, it is necessary to introduce multiple strain gauges, with one strain gauge placed at each force location. If superposition is used, which states that the total effect of multiple forces on a system is equal to the sum of the effects of each individual force, Equation (Equation 1) can be turned into a summation of all strain gauges to cover all the forces [11]. This is represented in Equation (Equation 7), where *m* represents the location at which the moment is calculated, *N* represents the total number of forces, and *n* is the dummy index. $L_{n}$ and $L_{m}$ represent the location of the forces. Lastly, the $F_{n}$ represents the force magnitude for various locations.
(7)$$M_{m}=\sum_{n=1}^{m}F_{n}L_{n}\left(1-\frac{L_{m}}{L}\right)+\sum_{n=m+1}^{N}F_{n}L_{m}\left(1-\frac{L_{n}}{L}\right)$$

In the experiment being conducted, seven force locations were utilized. If Equation (Equation 7) is to be written in its full matrix form to encompass all seven forces, it would result in Equation (Equation 8).
(8)$$\left[\begin{array}{c} M_{1}\\ M_{2}\\ M_{3}\\ M_{4}\\ M_{5}\\ M_{6}\\ M_{7} \end{array}\right]=\left[\begin{array}{ccccccc} L_{1}\left(1-\frac{L_{1}}{L}\right) & L_{1}\left(1-\frac{L_{2}}{L}\right) & L_{1}\left(1-\frac{L_{3}}{L}\right) & L_{1}\left(1-\frac{L_{4}}{L}\right) & L_{1}\left(1-\frac{L_{5}}{L}\right) & L_{1}\left(1-\frac{L_{6}}{L}\right) & L_{1}\left(1-\frac{L_{7}}{L}\right)\\ L_{1}\left(1-\frac{L_{2}}{L}\right) & L_{2}\left(1-\frac{L_{2}}{L}\right) & L_{2}\left(1-\frac{L_{3}}{L}\right) & L_{2}\left(1-\frac{L_{4}}{L}\right) & L_{2}\left(1-\frac{L_{5}}{L}\right) & L_{2}\left(1-\frac{L_{6}}{L}\right) & L_{2}\left(1-\frac{L_{7}}{L}\right)\\ L_{1}\left(1-\frac{L_{3}}{L}\right) & L_{2}\left(1-\frac{L_{3}}{L}\right) & L_{3}\left(1-\frac{L_{3}}{L}\right) & L_{3}\left(1-\frac{L_{4}}{L}\right) & L_{3}\left(1-\frac{L_{5}}{L}\right) & L_{3}\left(1-\frac{L_{6}}{L}\right) & L_{3}\left(1-\frac{L_{7}}{L}\right)\\ L_{1}\left(1-\frac{L_{4}}{L}\right) & L_{2}\left(1-\frac{L_{4}}{L}\right) & L_{3}\left(1-\frac{L_{4}}{L}\right) & L_{4}\left(1-\frac{L_{4}}{L}\right) & L_{4}\left(1-\frac{L_{5}}{L}\right) & L_{4}\left(1-\frac{L_{6}}{L}\right) & L_{4}\left(1-\frac{L_{7}}{L}\right)\\ L_{1}\left(1-\frac{L_{5}}{L}\right) & L_{2}\left(1-\frac{L_{5}}{L}\right) & L_{3}\left(1-\frac{L_{5}}{L}\right) & L_{4}\left(1-\frac{L_{5}}{L}\right) & L_{5}\left(1-\frac{L_{5}}{L}\right) & L_{5}\left(1-\frac{L_{6}}{L}\right) & L_{5}\left(1-\frac{L_{7}}{L}\right)\\ L_{1}\left(1-\frac{L_{6}}{L}\right) & L_{2}\left(1-\frac{L_{6}}{L}\right) & L_{3}\left(1-\frac{L_{6}}{L}\right) & L_{4}\left(1-\frac{L_{6}}{L}\right) & L_{5}\left(1-\frac{L_{6}}{L}\right) & L_{6}\left(1-\frac{L_{6}}{L}\right) & L_{6}\left(1-\frac{L_{7}}{L}\right)\\ L_{1}\left(1-\frac{L_{7}}{L}\right) & L_{2}\left(1-\frac{L_{7}}{L}\right) & L_{3}\left(1-\frac{L_{7}}{L}\right) & L_{4}\left(1-\frac{L_{7}}{L}\right) & L_{5}\left(1-\frac{L_{7}}{L}\right) & L_{6}\left(1-\frac{L_{7}}{L}\right) & L_{7}\left(1-\frac{L_{7}}{L}\right) \end{array}\right]\left[\begin{array}{c} F_{1}\\ F_{2}\\ F_{3}\\ F_{4}\\ F_{5}\\ F_{6}\\ F_{7} \end{array}\right]$$

Equation (Equation 8) can be written in a condensed form, represented by Equation (Equation 9), where [M] is the column vector representing the bending moment at each location, [F] is the column vector representing the magnitude of the applied force at each location, and [L] is the geometric matrix.
(9)$$\left[M\right]=\left[L\right]\left[F\right]$$

As matrix [L] is symmetric, it can be inverted to calculate the forces at each of the locations as represented in Equation (Equation 10). Additionally, Equation (Equation 6) can be used to replace the moment matrix with the strain gauge values from the experiment and multiplied by the calibration factor, as represented in Equation (Equation 11). This establishes the final relationship between the strain gauge readings and the forces.
(10)$$\left[F\right]={\left[L\right]}^{-1}\left[M\right]$$
(11)$$\left[F\right]=\beta {\left[L\right]}^{-1}\left[\varepsilon \right]$$

If the position of a force between two strain gauges is known, the force will be distributed to the two nearby strain gauges as shown in Figure 4. Force $F_{b}$ is the actual force being applied; however, the strain gauges will see that the force is distributed among the two strain gauges nearby as $F_{1}$ and $F_{2}$. $\ell_{b}$ represents the position of the force referenced from the left adjacent strain gauge and *ℓ* refers to the spacing between such adjacent strain gauges. In Equations (Equation 12) and (Equation 13), the forces at the adjacent strain gauges can be calculated by knowing such values.
(12)$$F_{1}=\left(1-\frac{\ell_{b}}{\ell }\right)F_{b}$$
(13)$$F_{2}=\frac{\ell_{b}}{\ell }F_{b}$$

## 3. Experimental Results

### 3.1. Experimental Setup

The experiments conducted here represent only a few of the many scenarios in which the methodology stated can be used for load identification purposes. The rationale behind selecting these scenarios is rooted in their prevalence within real-world situations. They represent commonly encountered cases rather than extreme or overly simplistic scenarios. As shown in Figure 5, masses were hung from a thin cord on top of a strain gauge or between strain gauges to simulate a point load condition. This was repeated for various placements of one or multiple loads on the beam. With the knowledge of the strain gauge locations, the strain readings, and the locations of the applied loads, one could use the methodology presented to calculate the force magnitudes. In order to conduct the experiment, certain values were recorded such as the strain gauge readings, the location of the strain gauges themselves, and the positions of the applied loads.

The setup for this experiment consisted of an aluminum 6061-T6 simply supported beam with the dimensions of 3.81 cm × 0.476 cm × 61 cm shown in Figure 5. At each end, 5.5 cm of the beam was in contact with the supports, resulting in a gap of 50 cm. Seven Vishay, uniaxial strain gauges of model number CEA-12-240-UZ-120 were attached to the underside of the beam, as seen in Figure 6, at locations shown in Figure 7. These strain gauges were applied to the beam by using Vishay application techniques [12]. Wires were soldered to the leads on the strain gauge to read the strain values by using two Vishay P3 indicators.

To establish the necessary connections, a total of three wires were used for each strain gauge. Referring to Figure 6, a red wire was connected to one of the pads on the strain gauge, while the second pad had two wires connected to it: white and black. Figure 8 illustrates the other end of these wires. The red wire was linked to the terminal labeled P+, the white wire to the terminal labeled S−, and finally, the black wire to the terminal labeled D120. *P* is the excitation voltage (typically 10 V) and *S* is the measurement voltage of the Wheatstone bridge. The connection of the black wire to the D120 terminal was based on the grid resistance of 120 ohms. This resistance value is determined by the strain gauge manufacturer and the construction of the strain gauge itself.

### 3.2. Calibration Factor

To effectively carry out the experiment, it was crucial to first determine the calibration factor. To obtain the value, a simultaneous uniform loading was applied. This was done by loading 10.8 N first on the center strain gauge and repeatedly adding 10.8 N adjacent to the existing loads until all gauges had a force applied. Upon collecting the data, the solver function in Excel was employed to attain the optimal fit for the force, ensuring it equaled 10.8 N. The actual calibration factor was determined to be 0.00932, and to verify its accuracy, the theoretical beta factor was calculated using the beam geometry and material properties from Equation (Equation 5), resulting in a value of 0.00994. The comparison between the theoretical beta and the calibrated beta value led to a minor difference of 6.3%, as shown in Table 1. By multiplying the obtained strain data by the calibration factor (beta), the moment could be calculated, providing insights into how the beam responds under different loading conditions.

### 3.3. Single Load Applied on Top of a Strain Gauge

For this condition, a 9.8 N load was applied on top of a single strain gauge and all the strains were read and shown in Table 2. For each test, the load was moved to a different strain gauge. The point location in Table 2 refers to the distance away from the left end at which the beam makes contact with the supports.

The moment graph was created by using Equation (Equation 6). By observing the moment graph for the 9.8 N experiment shown in Figure 9, it can be determined how each load applied affects the surrounding area on the beam. Wherever the load is applied on the beam, the moment distribution will peak at that location. This shows the concentration of stress and bending at that particular location where the load was applied.

The forces were calculated by taking the moment matrix and multiplying it by the inverse geometric matrix using Equation (Equation 11), and the results are presented in Table 3. Upon examining the table, it can be observed that the force at the strain gauge location with the applied load should be close to 9.8 N, while all other strain gauge locations should theoretically register zero force. This observation is consistent with the expectation that only the applied load should affect the force at the specific strain gauge location.

The force errors in the experiment take into account the differences between expected and calculated forces. In contrast, the zero-force errors refer to the comparison of forces at all other strain gauge locations without the applied force, effectively evaluating the accuracy of the zero-force assumption at these locations. To compute the averages for both force errors and zero force errors, the respective values were summed up and then divided by the total number of forces applied.

By comparing calculated and expected forces, average and zero errors can be computed not only for the 9.8 N experiment but also for the 19.6 N experiment. The results of this comparison can be observed in Table 4. This additional analysis helps to assess the overall accuracy and reliability of the experimental setup and the underlying mathematical models used for force calculation across different load magnitudes.

### 3.4. Multiple Loads Applied on Top of Strain Gauges

The adjacent load experiment (Figure 10) consisted of two 9.8 N forces applied at the first and second points, and then moving the pair of loads down the beam. Table 5 and Table 6 show the strain readings and calculated forces for this scenario simulating a two-axle vehicle moving along a bridge.

The skip strain gauge experiment (Figure 11) involved the application of two 9.8 N loads on strain gauges placed at specific locations, with an unloaded strain gauge in-between. The first load was placed on strain gauge 1 and the second load on strain gauge 3, intentionally skipping strain gauge 2. This trial skips strain gauge 2 which simulates loads being apart from each other with the strain gauge in the middle. This experiment is done to see how the beam responds to multiple loads that are not directly adjacent to each other. This same procedure is repeated for all the points. Table 7 and Table 8 show the strain readings and the calculated forces from this method.

In both multiple load experiments, the average force and zero error are calculated by comparing the forces with the expected value, and the results are presented in Table 9.

### 3.5. Single Load Applied In-Between Two Strain Gauges

The data collected from the in-between test used a 19.6 N load which was applied at various different ratios between known locations. Three different ratios were tested, 25%, 50%, and 75%. The load distribution relies solely on the location of the load relative to the adjacent strain gauges. For example, if the load is applied 25% away from strain gauge 2, that means the force is 75% away from strain gauge 3. Inversely, the experienced load at strain gauge 2 will be 75% of the applied load, and strain gauge 3 will experience 25% of the applied load. Table 10 shows the data that were collected for this experiment. From the data collected, the data points start at strain gauge 3, where 3.25 represents the load 25% of the way of the strain gauge, 3.50 represents the load 50% between the strain gauges 3 and 4, and 3.75 represents the load being applied 75% of the way between the strain gauges 3 and 4. The final forces that were calculated are shown in Table 11.

Figure 12 shows the force distribution for the in-between test. It relates to the values from Table 11, solidifying the concept of the method by showing that placing a load 50% away from one strain gauge and 50% away from the other indicates that both of the strain gauges should experience a similar amount of load. Keep in mind that this is not the full magnitude of the force but rather 50% as the force was placed directly in the middle. Figure 13 shows the specific case of the load being placed halfway between strain gauges 4 and 5. Theoretically, the peak of the force should occur directly between strain gauges 4 and 5. As there is no strain gauge available in this scenario, to get the peak force value, it can be observed that the force is distributed into strain gauges 4 and 5 shown by the orange line. Referring back to Figure 4, this is the expected outcome. Where the applied force $F_{b}$ cannot be directly measured, the strain gauges pick up the force that is now distributed by the beam, and the distance is what determines the magnitude of the force that will be seen by the strain gauges, and this is supported by Equations (Equation 12) and (Equation 13). This can be seen in Figure 13, by drawing a theoretical dotted line that peaks between strain gauges 4 and 5, as represented by the blue dotted line. Using Table 11, the average force error and the zero force error can now be calculated which will determine how accurate this experiment was in determining the magnitude of a force that was not directly atop a strain gauge; the results are shown in Table 12.

### 3.6. Load Applied on Top of All Strain Gauges

The experiment evaluated the system response under simultaneous loading, exploring varied and uniform weight conditions. The varied case was initially tested by applying distinct loads to each strain gauge to validate the ability of the system to distinguish the loads based on their magnitudes. Table 13 outlines the tested scenarios for this case, beginning with a single gauge under load and incrementally adding loads to the remaining gauges. The recorded strain readings from the experiment are documented in Table 14. A moment diagram was subsequently derived using the above equations, as illustrated in Figure 14.

The calculated moments were then used to estimate the forces. As indicated in Table 15, the forces were largely accurate in identifying correct magnitudes, barring the final scenario where a load was applied at every gauge, including the ends. It is postulated that the errors in force readings arose from the lifting off of the edges of the beam from the surfaces they were resting on, leading to inaccurate strain readings.

The second tested case was the simultaneous uniform case, where identical loads were applied to all strain gauges. Table 16 presents the tested scenarios for this case. Strain values were recorded during the experiment, as shown in Table 17. A moment diagram was subsequently derived, with the moment curve represented in Figure 15. Interestingly, this curve exhibited a single peak despite identical loads being applied at every strain gauge—a result of the superposition principle, where the individual load peaks accumulate into a larger single peak.

The moments were then utilized to estimate the forces, as detailed in Table 18. The forces were plotted to examine their peaks, as illustrated in Figure 16. Despite the moment curve displaying a single peak, the superposition principle allowed the discernment of individual force magnitudes. The final scenario, with an identical load applied at all gauges, yielded a nearly flat line at the top, indicating the uniformly applied force.

Upon calculating the forces, force averages were determined to gauge the accuracy of both cases, as shown in Table 19. Two primary distinctions between the cases and their force average errors emerged. The first potential cause is the calibration factor, originally derived from a case similar to the uniform load scenario, which may account for its lower error. Additionally, the edges of the beam were observed to lift from the surfaces they were resting on during the varied load case, potentially contributing to additional errors. Despite these challenges, both cases were successful overall in accurately identifying the force loads applied to the beam.

## 4. Conclusions

This new methodology of force determination using a uniaxial strain gauge at each force location yielded results such that the process can be deemed accurate for simply supported, uniform beams. Additionally, this methodology identifies the magnitudes of loads at given locations while successfully calculating zero force loads elsewhere on the beam. The major benefit of this methodology is the minimization of the number of strain gauges required to determine force magnitudes on a beam. A reduction in strain gauges minimizes the number of channels required, thus further lowering the overall cost per test.

Before the experimental data were collected, the goal was to have the error fall within 5%. The overall force errors for cases where the loads were in line with the strain gauge locations peaked at 4.9%. The majority of errors for these data sets could be attributed to slight discrepancies in strain gauge placements and load location, as well as small imperfections in the thickness of the beam. Additionally, movement could have occurred resulting in the designated resting areas possibly shifting.

The data from the load being applied between strain gauges yielded an error of 6.6%. This is slightly higher than the goal, but causes for errors are most likely attributed to the inaccurate placement of the loads between strain gauges. Overall, the methodology and theory both have the ability to be modified for various load applications to suit any need while maintaining accuracy.

## Figures and Tables

**Figure 1 sensors-23-07017-f001:**
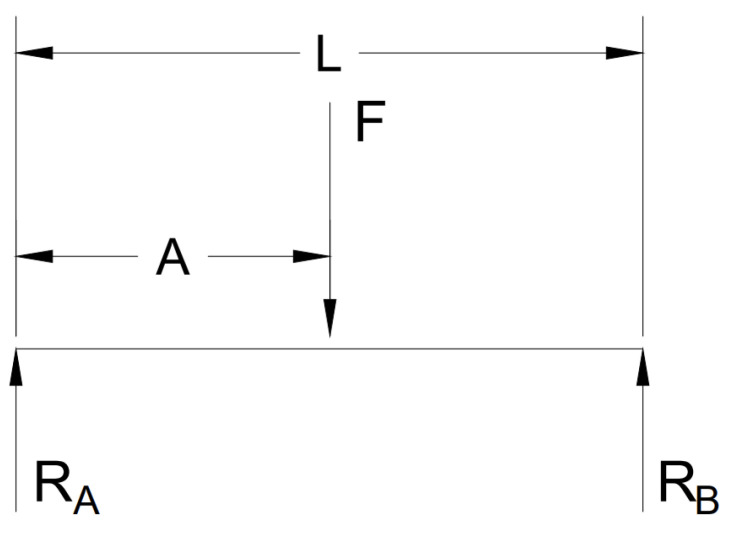
Simply Supported Beam with Single Point Force.

**Figure 2 sensors-23-07017-f002:**
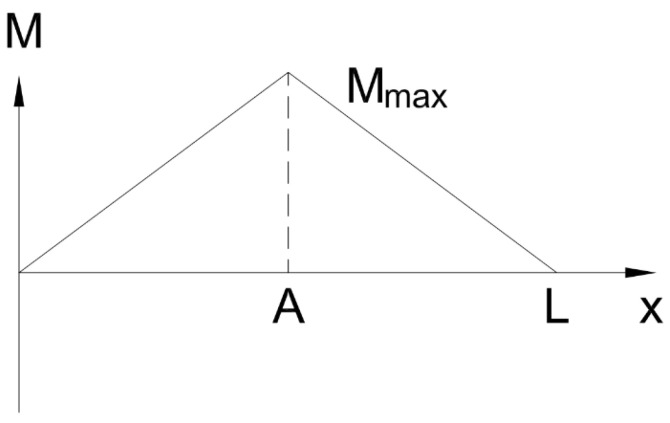
Bending Moment Curve for Simply Supported Beam with Single Point Force.

**Figure 3 sensors-23-07017-f003:**
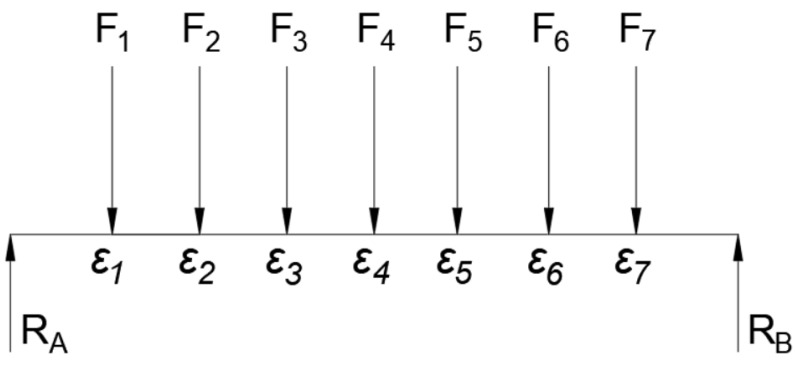
Strain Gauge Locations and Corresponding Force Labels for Reference.

**Figure 4 sensors-23-07017-f004:**
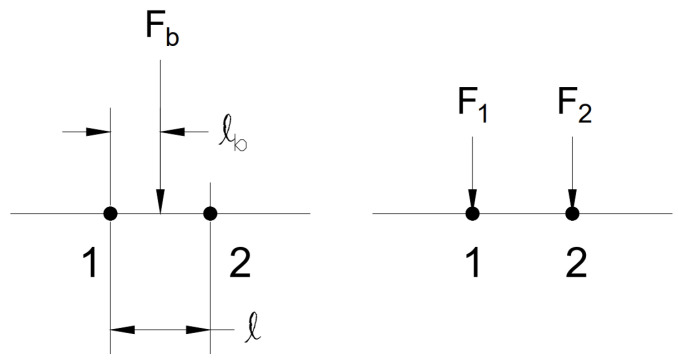
Theoretical Force Distribution Between Adjacent Strain Gauges in In-Between Case.

**Figure 5 sensors-23-07017-f005:**
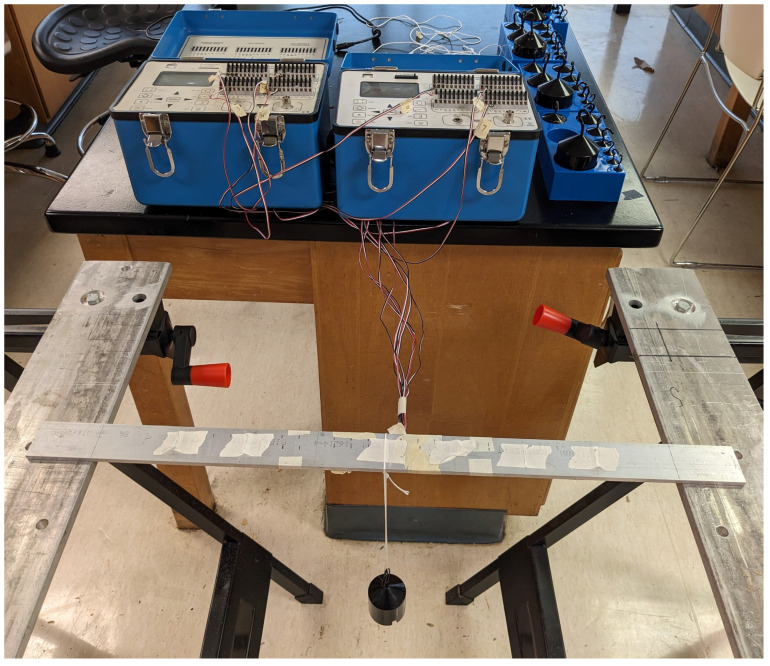
Experimental Setup.

**Figure 6 sensors-23-07017-f006:**
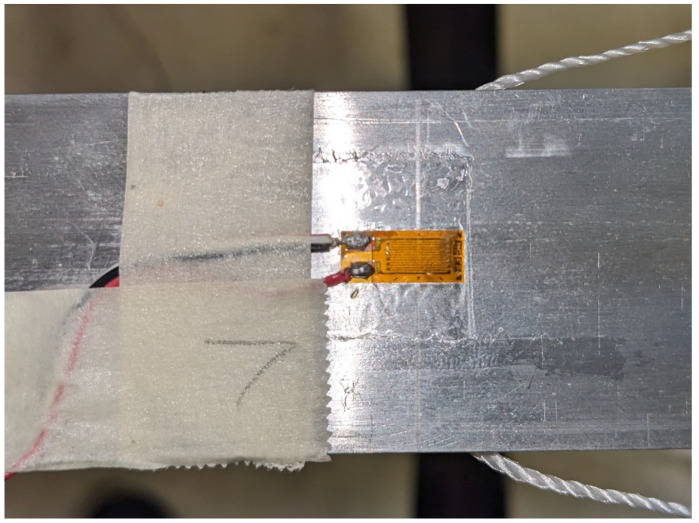
Installation of Strain Gauges on the Beam.

**Figure 7 sensors-23-07017-f007:**
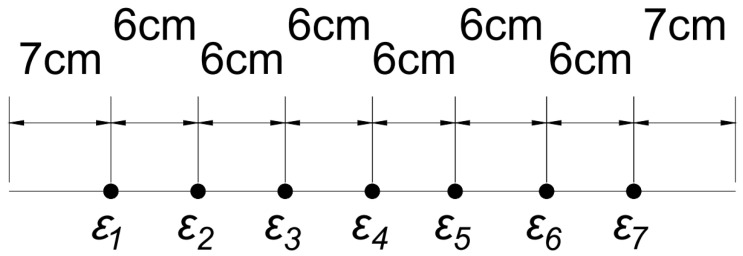
Beam Layout with Strain Gauge Locations and Dimensions.

**Figure 8 sensors-23-07017-f008:**
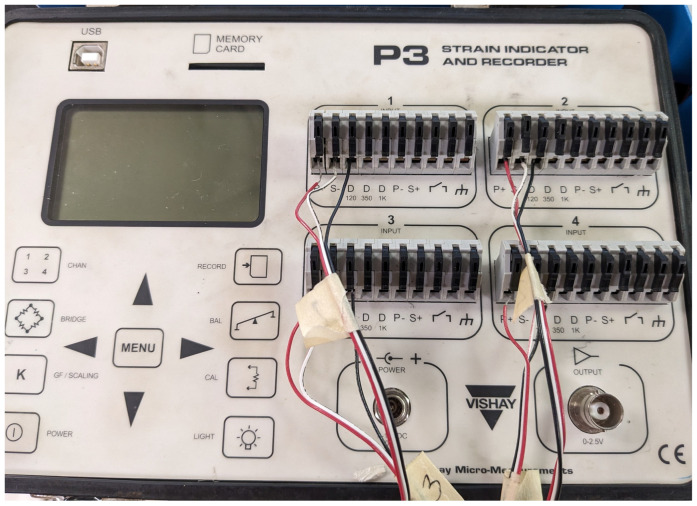
Vishay P3 Strain Indicator and Recorder.

**Figure 9 sensors-23-07017-f009:**
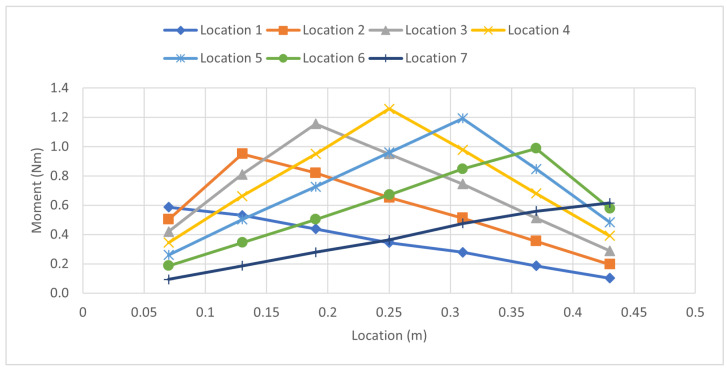
Moment Distribution: Single Load 9.8 N Case.

**Figure 10 sensors-23-07017-f010:**
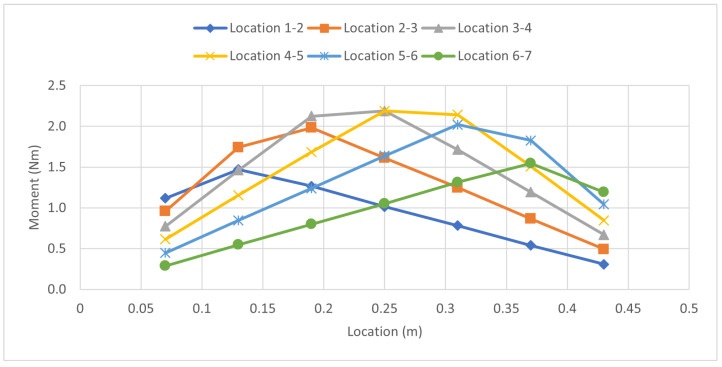
Moment Distribution: Adjacent Load Case.

**Figure 11 sensors-23-07017-f011:**
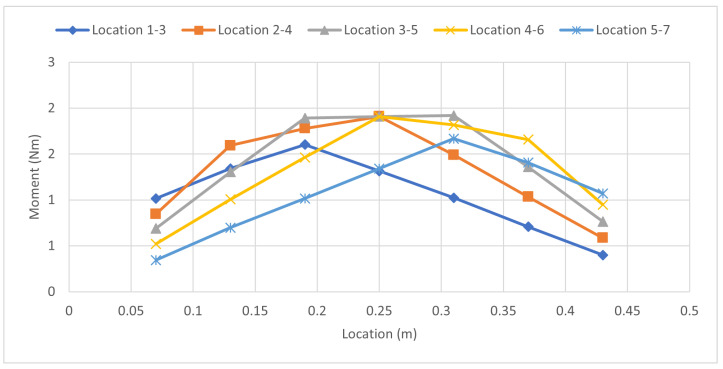
Moment Distribution: Skip Strain Gauge Load Case.

**Figure 12 sensors-23-07017-f012:**
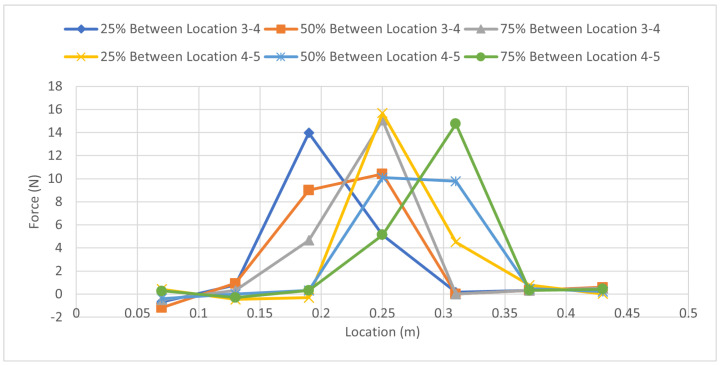
Force Distribution: In-Between Load 44.5 N Case.

**Figure 13 sensors-23-07017-f013:**
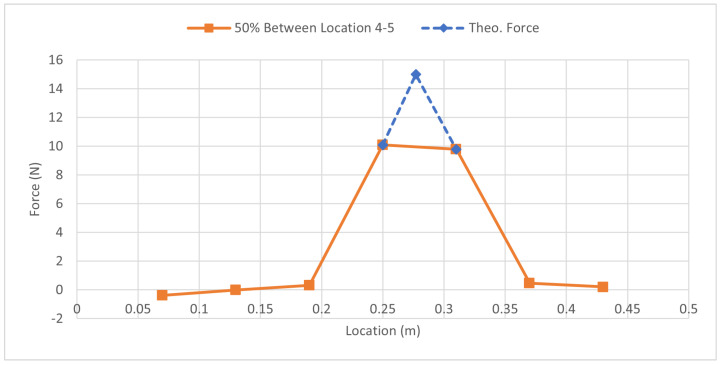
Comparison of Force Distribution Curve for 50% Load Between Strain Gauges 4 and 5 and Theoretical Applied Load.

**Figure 14 sensors-23-07017-f014:**
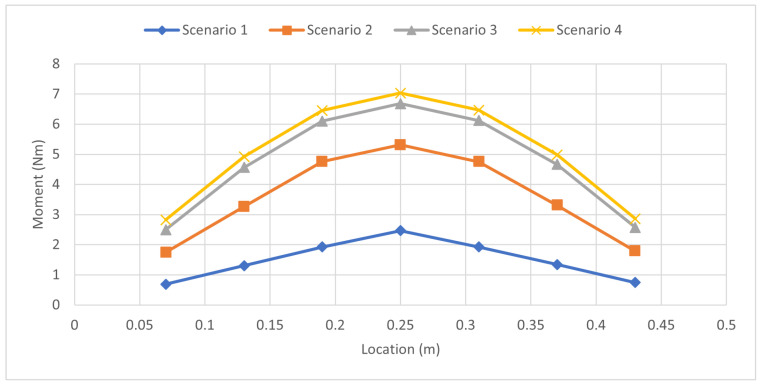
Moment Distribution: Simultaneous Varied Load Case.

**Figure 15 sensors-23-07017-f015:**
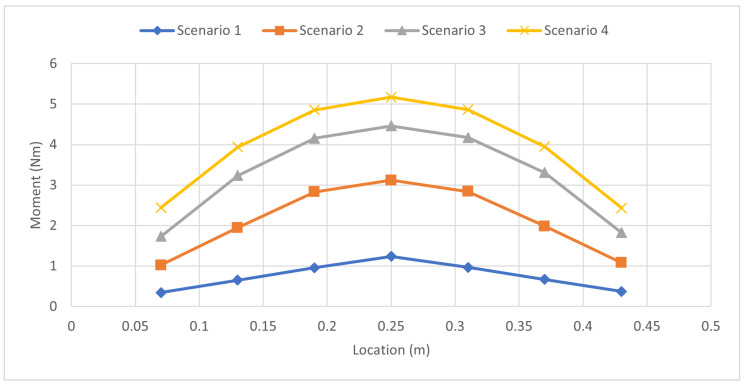
Moment Distribution: Simultaneous Uniform Load Case.

**Figure 16 sensors-23-07017-f016:**
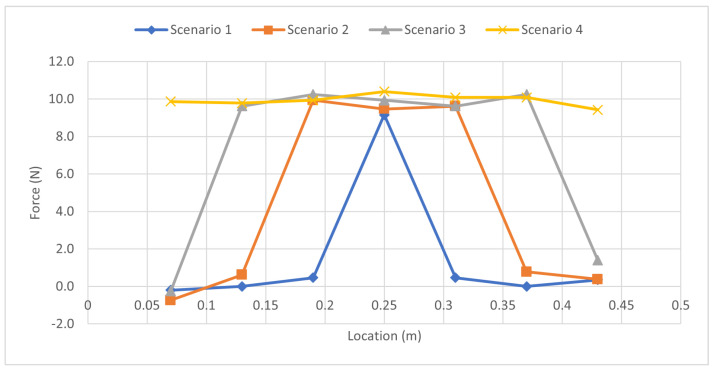
Force Distribution: Simultaneous Uniform Load Case.

**Table 1 sensors-23-07017-t001:** Beta Calculation Values.

b (m)	h (m)	I (m^4^)	y (m)	E (Pa)	Theo. Beta	Calibrated Beta	% Difference
0.0381	0.00476	3.43 × 10^−10^	0.00238	69 × 10^9^	0.00994	0.00932	6.3%

**Table 2 sensors-23-07017-t002:** Strain Readings: Single Load 9.8 N Case.

Point (m)	Strain (με)						
	1	2	3	4	5	6	7
0.07	63	57	47	37	30	20	11
0.13	54	102	88	70	55	38	21
0.19	45	87	124	102	80	55	31
0.25	37	71	102	135	105	73	42
0.31	28	54	78	103	128	91	52
0.37	20	37	54	72	91	106	62
0.43	10	20	30	39	51	60	66

**Table 3 sensors-23-07017-t003:** Force Values: Single Load 9.8 N Case.

Point (m)	Force (N)						
	1	2	3	4	5	6	7
0.07	9.3	0.6	0.0	−0.5	0.5	−0.2	0.1
0.13	−0.3	9.6	0.6	−0.5	0.3	0.0	0.2
0.19	−0.5	0.8	9.2	0.0	0.5	−0.2	0.4
0.25	−0.4	0.5	−0.3	9.8	0.3	−0.2	0.8
0.31	−0.3	0.3	−0.2	0.0	9.6	0.3	0.9
0.37	0.0	0.0	−0.2	−0.2	0.6	9.2	1.4
0.43	−0.2	0.0	0.2	−0.5	0.5	0.5	9.7

**Table 4 sensors-23-07017-t004:** Error Values: Single Loading Cases.

	9.8 N	19.6 N
Force Average Error	3.3%	4.9%
Zero Average Error	3.5%	3.1%

**Table 5 sensors-23-07017-t005:** Strain Readings: Adjacent Load Case (2 × 9.8 N).

Point 1 (m)	Point 2 (m)	Strain (με)						
		1	2	3	4	5	6	7
0.07	0.13	120	158	136	109	84	58	33
0.13	0.19	103	187	213	173	134	93	53
0.19	0.25	83	157	228	235	184	128	72
0.25	0.31	66	124	181	235	230	162	91
0.31	0.37	48	91	133	176	217	196	112
0.37	0.43	31	59	86	113	141	166	128

**Table 6 sensors-23-07017-t006:** Force Values: Adjacent Load Case (2 × 9.8 N).

Point 1 (m)	Point 2 (m)	Force (N)						
		1	2	3	4	5	6	7
0.07	0.13	10.1	9.3	0.8	−0.3	0.2	−0.2	0.5
0.13	0.19	0.7	9.0	10.2	−0.2	0.3	−0.2	0.8
0.19	0.25	−0.4	0.5	9.9	9.0	0.8	0.0	0.9
0.25	0.31	−0.2	0.2	0.5	9.2	9.8	0.5	1.1
0.31	0.37	−0.3	0.2	−0.2	0.3	9.6	9.8	1.9
0.37	0.43	−0.2	0.2	0.0	−0.2	0.5	9.8	11.1

**Table 7 sensors-23-07017-t007:** Strain Readings: Skip Strain Gauge Load Case (2 × 9.8 N).

Point 1 (m)	Point 2 (m)	Strain (με)						
		1	2	3	4	5	6	7
0.07	0.19	109	144	172	141	110	76	43
0.13	0.25	91	171	191	205	160	111	63
0.19	0.31	74	140	203	205	206	146	82
0.25	0.37	56	108	157	205	195	178	102
0.31	0.43	37	75	109	144	179	151	115

**Table 8 sensors-23-07017-t008:** Force Values: Skip Strain Gauge Load Case (2 × 9.8 N).

Point 1 (m)	Point 2 (m)	Force (N)						
		1	2	3	4	5	6	7
0.07	0.19	9.1	1.1	9.2	0.0	0.5	−0.2	0.6
0.13	0.25	−0.3	9.3	0.9	9.2	0.6	−0.2	0.9
0.19	0.31	−0.4	0.5	9.5	0.2	9.5	0.6	1.0
0.25	0.37	−0.6	0.5	0.2	9.0	1.1	9.2	1.8
0.31	0.43	−1.0	0.6	−0.2	0.0	9.8	1.2	9.7

**Table 9 sensors-23-07017-t009:** Average Error Values: Multiple Loading Cases (Adjacent and Skip Strain Gauge).

	Adjacent	Skip Strain Gauge
Force Average Error	4.4%	4.9%
Zero Average Error	4.3%	6.1%

**Table 10 sensors-23-07017-t010:** Strain Readings: In-Between Load 19.6 N Case.

Location	Strain (με)						
	1	2	3	4	5	6	7
3.25	86	164	237	220	170	119	66
3.50	80	156	226	238	183	128	71
3.75	78	148	216	254	195	136	75
4.25	72	131	193	257	220	154	83
4.50	66	125	184	241	233	162	88
4.75	64	117	172	225	245	170	93

**Table 11 sensors-23-07017-t011:** Force Values: In-Between Load 19.6 N Case.

Location	Force (N)						
	1	2	3	4	5	6	7
3.25	−0.7	0.8	14.0	5.1	0.2	0.3	0.6
3.50	−1.2	0.9	9.0	10.4	0.0	0.3	0.6
3.75	−0.5	0.3	4.7	15.1	0.0	0.3	0.5
4.25	0.4	−0.5	−0.3	15.7	4.5	0.8	0.0
4.50	−0.4	0.0	0.3	10.1	9.8	0.5	0.2
4.75	0.3	−0.3	0.3	5.1	14.8	0.3	0.4

**Table 12 sensors-23-07017-t012:** Average Error Values: In-Between Load 44.5 N Case.

	In-Between Strain Gauges
Force Average Error	4.5%
Zero Average Error	2.1%

**Table 13 sensors-23-07017-t013:** Applied Forces to Each Strain Gauge in the ’Simultaneous Varied Load Case’ Experiment.

Scenarios	Strain Gauge						
	1	2	3	4	5	6	7
1	0	0	0	19.6 N	0	0	0
2	0	0	14.7 N	19.6 N	14.7 N	0	0
3	0	9.8 N	14.7 N	19.6 N	14.7 N	9.8 N	0
4	4.9 N	9.8 N	14.7 N	19.6 N	14.7 N	9.8 N	4.9 N

**Table 14 sensors-23-07017-t014:** Strain Readings: Simultaneous Varied Load Case.

Point (m)	Strain (με)			
	1	2	3	4
0.07	74	187	268	303
0.13	140	350	490	528
0.19	207	511	655	693
0.25	265	570	717	755
0.31	207	510	657	694
0.37	144	355	501	535
0.43	80	192	276	307

**Table 15 sensors-23-07017-t015:** Force Values: Simultaneous Varied Load Case.

Point (m)	Force (N)			
	1	2	3	4
0.07	−0.4	−0.4	1.2	5.4
0.13	−0.2	0.3	8.9	9.3
0.19	1.4	15.8	16.0	16.0
0.25	18.0	18.5	18.9	19.1
0.31	0.8	14.8	14.9	15.2
0.37	0.2	1.2	10.7	10.7
0.43	0.7	0.2	1.8	5.5

**Table 16 sensors-23-07017-t016:** Applied Forces to Each Strain Gauge in the ’Simultaneous Uniform Load Case’ Experiment.

Scenarios	Strain Gauge						
	1	2	3	4	5	6	7
1	0	0	0	9.8 N	0	0	0
2	0	0	9.8 N	9.8 N	9.8 N	0	0
3	0	9.8 N	9.8 N	9.8 N	9.8 N	9.8 N	0
4	9.8 N	9.8 N	9.8 N	9.8 N	9.8 N	9.8 N	9.8 N

**Table 17 sensors-23-07017-t017:** Strain Readings: Simultaneous Uniform Load Case.

Point (m)	Strain (με)			
	1	2	3	4
0.07	37	110	186	262
0.13	70	209	347	423
0.19	103	304	446	521
0.25	133	335	479	555
0.31	104	305	448	522
0.37	72	213	355	424
0.43	40	116	196	261

**Table 18 sensors-23-07017-t018:** Force Values: Simultaneous Uniform Load Case.

Point (m)	Force (N)			
	1	2	3	4
0.07	−0.2	−0.7	−0.2	9.9
0.13	0.0	0.6	9.6	9.8
0.19	0.5	9.9	10.2	9.9
0.25	9.2	9.5	9.9	10.4
0.31	0.5	9.6	9.6	10.1
0.37	0.0	0.8	10.2	10.1
0.43	0.4	0.4	1.4	9.4

**Table 19 sensors-23-07017-t019:** Error Values: Simultaneous Load Case.

	Varied	Uniform
Force Average Error	6.5%	2.8%
Zero Average Error	10.9%	4.8%

## Data Availability

Data is contained within the article.

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
