# Peer review of "Determining Magnitudes of Forces at Known Locations through a Strain Gauge Force Transducer"

_sensors, 2023, doi:10.3390/s23167017_

Round 1
Reviewer 1 Report
The paper innovatively developed a train gauge force transducer to minimize the number of strain gauges needed to determine the magnitudes of loads when the locations are known. Regarding to the quality and presentation of this paper, I would like to recommend a MAJOR REVISION of it.
1. However, the structure of the paper can be improved. Some parts of writing can be clearer. Discussion can be in more depth.
2. Some important related references are suggested to be cited in the manuscript.
[1] Liang, Qiaokang, et al. "A Force/Position Measurement Method of Surgical Forceps Based on Fiber Bragg Gratings." IEEE Sensors Journal 23.1 (2022): 363-373.
[2] Long, Jianyong, et al. "Ultrathin three-axis FBG wrist force sensor for collaborative robots." IEEE Transactions on Instrumentation and Measurement 70 (2021): 1-15.
[3] Wang, Qinghua, et al. "New Technique for Impact Calibration of Wide-Range Triaxial Force Transducer Using Hopkinson Bar." Sensors 22.13 (2022): 4885.
Reviewer 2 Report
This article presents a method for the determination of forces at known application points by means of a purposely developed, strain gauge-based, force transducer.
The reviewer's comments are reported below:
- The advantage of the proposed method is not clear. It is claimed that such a method might have several applications, however not a single practical example is provided.
- Why the authors used exactly 7 strain gauges? There is not explanation for such a quantity. The authors should also explain how many gauges would be required to perform the same experiments with traditional tachniques.
- Since knowledge of the location of the applied loads is mandatory to implement the proposed method, it is not clear the advantage with respect to other methods such as the one of [1] where, with two strain gagues, it is possible to measure the applied force without knowing the application point. In general, bibliography is poor.
- The experimental setup should be better constructed. In the Conclusions, authors correctly report several limitations that do lower the experiments reliability. Moreover, hanging the weights as it is illustrated in the article might lead to non-vertical load application.
Minor
- In the setup, cables and cords visible in Fig. 5 and Fig. 6 could be better arranged. Quality of such figures, especially of Fig. 7, must be improved.
- Still in the setup, some more information about the devices used for recording the gauges output (Vishay P3) should be given.
[1] R.A. Romeo et al., "Instrumenting a Robotic Finger to Augment the Capabilities of Robotic Grippers", IEEE Transactions on Instrumentation and Measurement, 2023.
Pleas avoid using Saxon Genitive with unanimated items.
Please would avoid using first persons ("we").
Tenses must be coherent throughout the whole text.
Reviewer 3 Report
The paper describes a force transducer utilizing strain-gauge technology to reduce the number of required strain gauges for determining load magnitudes when the locations are known. The topic of the paper is interesting, and the subject matter is well-executed.
I have carefully reviewed the content and findings presented in the paper and would like to provide some suggestions and comments.
· The references should be an integral part of the sentence (as done with references 1-6), while the other references are listed later after completing the sentence.
· At the beginning of Chapter 3, "Experimental Results," it is emphasized that only some of the possible scenarios were investigated in the study. Although the applied scenarios are described in detail later in the paper, it would be helpful to clarify immediately if these are the most specific and worst-case scenarios for the method's application and if covering only those scenarios was sufficient for the research.
Aluminum 6061-T6 was used as the beam material. The rationale for this choice should be explained, highlighting the main characteristics of the material that could affect the response of the strain gauges. Additionally, if possible, provide a link to the used strain gauges or emphasize their most significant characteristics in the paper.
· Figure 7 depicts the installation of the strain gauges. However, the details of the strain gauges are not clearly visible in the image. Therefore, I recommend either capturing an enlarged image focused solely on the strain gauges or providing a detailed drawn design.
· Grammar and spelling should be checked throughout the paper, such as in Chapter 3.2, the third sentence.
· Are the characteristics of every load measurement determined from a single measurement? It would be beneficial to verify the repeatability of the obtained values.
Round 2
Reviewer 1 Report
1. Some Figs are too big. The authors are suggested to redraw the Figs.
2. References in the manuscript is not enough. Some important recent references are missed.
3. As a suggestion, the author may point out only the most important achievements of the proposed method in the conclusion with a similar style.
Author Response
Authors: Thank you again for your follow-up comments. We have carefully considered your additional suggestions with detailed comments below.
- Some Figs are too big. The authors are suggested to redraw the Figs.
Authors: This was a very good suggestion. We have adjusted the figure sizes accordingly.
- References in the manuscript is not enough. Some important recent references are missed.
Authors: We have carefully considered the references you provided in the first round of comments and indeed were able to incorporate two of them. We believe that all relevant current references have been cited.
- As a suggestion, the author may point out only the most important achievements of the proposed method in the conclusion with a similar style.
Authors: Thank you greatly for this suggestion. The entire conclusion has been rewritten.
Reviewer 2 Report
In my previous report, I asked to clarify the advantage of the proposed method against traditional techniques employing strain gauges. In my opinion, it is not sufficient to mention reference [8].
Theoretically, it may be possible to use seven quarter-bdrige load cells, therefore with the same number of strain gagues as in the proposed method, and obtain similar results both when a single load is applied on the beam and when multiple loads are applied.
Therefore, to confirm the novelty of their approach, I would kindly ask the authors to provide evidence of the superiority of the proposed method with respect to using seven monoaxial, quarter-bridge load cells in place of the fixed strain gauges. If no experiments can be performed, at least a mathematical demonstration is required.
English was improved according to previous suggestions.
Author Response
Authors: Thank you again for your follow-up comments. We have carefully considered your additional suggestions with detailed comments below.
- In my previous report, I asked to clarify the advantage of the proposed method against traditional techniques employing strain gauges. In my opinion, it is not sufficient to mention reference [8].
Authors: Thank you for this observation. The main advantage of the methodology presented here is that it minimizes the amount of strain gauges needed to perform load identification on a beam. Reference [8] is relevant because that methodology uses more strain gauges than the one presented here.
- Theoretically, it may be possible to use seven quarter-bdrige load cells, therefore with the same number of strain gagues as in the proposed method, and obtain similar results both when a single load is applied on the beam and when multiple loads are applied.
Authors: You are correct that there may be some advantages to use load cells instead of fixed uniaxial strain gauges (portability being one). However, the goal of this methodology was to minimize cost. Uniaxial strain gauges are a much cheaper alternative then compressive load cells.
- Therefore, to confirm the novelty of their approach, I would kindly ask the authors to provide evidence of the superiority of the proposed method with respect to using seven monoaxial, quarter-bridge load cells in place of the fixed strain gauges. If no experiments can be performed, at least a mathematical demonstration is required.
Authors: Thank you again for your suggestion. Related to the comment above, a load cell basically incorporates a strain gauge (or multiple strain gauges) inside of the device, so in theory it should yield similar results. It would basically be the same as putting a series of scales underneath the loads. The novelty of this method is that the beam itself is turned into a transducer using the strain gauges.